# Meshing Anxiety, Depression, Quality of Life, and Functionality in Chronic Disease

**DOI:** 10.3390/healthcare13050539

**Published:** 2025-03-02

**Authors:** Ana Pedro Costa, Irma da Silva Brito, Teresa Dionísio Mestre, Ana Matos Pires, Manuel José Lopes

**Affiliations:** 1University of Évora, 7004-516 Évora, Portugal; tmestre82@gmail.com; 2Comprehensive Health Research Center, 1150-082 Lisboa, Portugal; ana.matos.pires@gmail.com (A.M.P.); mjl@uevora.pt (M.J.L.); 3Psychiatry Service of Mental Health Department, Health Unit of Baixo Alentejo (ULSBA), 7801-849 Beja, Portugal; 4Nursing School of Coimbra, 3004-011 Coimbra, Portugal; irmabrito@esenfc.pt; 5Health Sciences Research Unit: Nursing, Nursing School of Coimbra, 3046-851 Coimbra, Portugal; 6Department of Health, Polytechnic Institute of Beja, 7800-295 Beja, Portugal; 7São João de Deus School of Nursing, University of Évora, 7000-811 Évora, Portugal

**Keywords:** anxiety, depression, quality of life, functionality, chronic disease

## Abstract

Background/Objectives: Anxiety and depression result in a greater health burden; both can impact functionality and quality of life. This review aims to understand the association between anxiety, depression, functionality, and quality of life. Although three systematic reviews exist, one focuses on quality of life in depression and anxiety, while the others address functionality in depression and anxiety, with the former being more recent and the latter two being older. The association between these four variables will be explored. Methods: A literature search of MEDLINE with Full Text, CINHAL PLUS with Full Text, Psychology and Behavioral Sciences Collection, and Academic Search Complete was conducted from 1 January 2017 to 22 November 2022. Seven studies involving 2279 adults were included. Results: All studies analyzed the association between anxiety and/or depression with functionality and/or quality of life, in a population with a primary chronic condition. We found the higher functionality (return to work, no sedentary lifestyle, and no fatigue) and quality of life are, the lower the values of anxiety and depression will be. The HADS is a consensual instrument to access anxiety and depression, but the same cannot be said for assessing functionality and quality of life. Conclusions: The severity of the chronic disease and the loss of functionality and quality of life seem to increase psychological distress. This study highlights the importance of a multidisciplinary and holistic approach, focusing not only on clinical outcomes but also on overall well-being. Further longitudinal research is needed to support the association between these variables to draw more concrete conclusions with scientific evidence.

## 1. Introduction

Mental disorders are a key contributor to the global disease burden, ranking among the top causes for disability [1]. Anxiety and depression, which are the most prevalent (4.4% and 4% worldwide, respectively, in 2021), often result in a greater health burden (538.7 and 713.8 disability-adjusted life years [DALYs] per 100,000 people worldwide, respectively, in 2021) than serious mental illnesses [2]. Individuals with anxiety and depression may experience significant limitations in their functionality and a decline in their overall quality of life [1].

People with depressive disorders often share clinical features with those with anxiety disorders. In addition, depression is a frequent complication of anxiety disorders [3]. Both clinical entities can occur together, thus fulfilling the diagnostic criteria for both disorders [4]. It is sometimes difficult to distinguish between the two, but it is important to identify and address both [3]. Anxiety and depression have been associated with adverse social and individual effects, with higher health care costs [5,6] and an increased risk of comorbidities [7,8].

Depression disorders are characterized by a depressed mood, decreased interest, decreased pleasure, insomnia or hypersomnia, agitation or psychomotor slowing, fatigue or loss of energy, decreased ability to concentrate, thoughts of devaluation, and guilt or death [9]. Anxiety disorders are characterized by physical symptoms (a shortness of breath, lump in the throat, chills, tremors, and palpitations), and psychological symptoms (fear, uneasiness, discomfort, insecurity, and apprehension about the future) [10]. Depressive and anxiety disorders have an impact on people’s lives, especially on functioning. The Diagnostic and Statistical Manual of Mental Disorders, Fifth Edition, Text Revision [DSM-5-TR] diagnostic criteria [9] detail these symptoms, but also address personal perceptions of quality of life, i.e., how the individual evaluates personal satisfaction in different dimensions [11].

The World Health Organization [WHO] defines functionality as a “dynamic interaction between a person’s health condition, environmental factors and personal factors” and recognizes that experiencing restrictions in participation leads to disability [12]. Functionality should be assessed according to what the individual does daily, taking in account the resources available, and is related to physical and/or mental factors and factors of a social, economic, or environmental nature [13]. Patients with anxiety and depression frequently experience impairment in several functional areas, including their capacity for everyday activities, for establishing and maintaining interpersonal relationships, for productivity at work, and in physical functionality [14,15]. McKnight et al. (2009) carried out a systematic review that shows the link between depressive symptom severity and functional impairment to be often bidirectional [16]. McKnight et al. (2016) conducted a systematic review that showed a correlation between symptoms of anxiety disorders across functional domains (global, social, occupational, and physical) [14].

The WHO defined quality of life as “an individual’s perception of their position in life, in the context of the culture and value systems in which they live and in relation to their goals, expectations, standards, and concerns” [17]. Thus, quality of life is seen as a cross-cultural and multidimensional concept, which is influenced in a complex way by the individual’s physical and mental health, level of independence, social relationships, personal beliefs, and relationships with the relevant characteristics of their environment [18]. Hohls et al. (2021) recently conducted a systematic review that showed that quality of life is lower before the development of anxiety and depressive disorders, reduces further at the onset of these disorders, and generally improves with remission to premorbid levels [19]. The results show a decrease in quality of life even before the onset of these disorders, highlighting the importance of early preventive measures in vulnerable groups. The course of the disorder (recurrent, relapsing, or chronic) seems to play an important role on quality of life as well. In addition, this review also reports on a bidirectional effect, where quality of life is predictive of mental health outcomes [19].

Chronic diseases significantly affect mental health, posing an ongoing challenge in public health [20]. Chronic diseases are long-term conditions not caused by infections or contact, often resulting from harmful behaviors and socioenvironmental factors [21]. The main chronic diseases include cardiovascular disease, cancer, chronic respiratory diseases, and diabetes [21], which account for most related deaths, and significantly contribute to the global disease burden [22]. These can interrupt an individual’s normal activities and functionality, causing a poorer quality of life [23]. There is a mutual influence between chronic illnesses and mental health disorders; chronic diseases can lead to emotional distress, anxiety, and depression [24], with a bidirectional association [25]. The demands of managing chronic illnesses often lead to emotional distress, while mental health problems are linked to unhealthy habits, reduced adherence to treatments, and impaired immune function [20].

Health services face a paradigm, unlike in the past, when the objective was a direct decrease in symptoms. Nowadays, the approach is more comprehensive and holistic, also considering the quality of life and functionality of patients [26]. Thus, a greater emphasis is placed on recovery, with a focus on rehabilitation, so both quality of life and functionality are increasingly becoming key factors in the treatment of both mental illness and chronic diseases [23,26].

Even though there are already three systematics reviews [14,16,19], one only assesses quality of life in depression and anxiety and the others assess functionality in depression and functionality in anxiety, with the first being a more recent revision (2021) and the other two being older (2009, 2016). Thus, the association between these four variables can be further explored. Considering the high prevalence of these disorders and their impact on both quality of life and functionality, there is a need to mesh those variables in a practice approach. The aim of this systematic review is to analyze the association between anxiety, depression, functionality, and quality of life.

## 2. Materials and Methods

### 2.1. Search Strategy and Selection Criteria

This systematic review was carried out using the Preferred Reporting Items for Systematic Reviews and Meta-Analyses (PRISMA) reporting guidelines [27]. The protocol was registered on PROSPERO (CRD42022303760); however, some adjustments were conducted to have enough articles to include in the review: the electronic databases and the inclusion and the exclusion criteria were changed, leading to the inclusion of studies with primary chronic conditions.

The National Library of Medicine, through PubMed, was searched for MeSH words. The search was conducted for studies published in English, with full text available, between 1 January 2017 and 22 November 2022. In addition, we searched the following databases for the terms (anxiety OR anxiety disorders) AND (depressive disorder OR depression OR depression disorder, major) AND (international classification of functioning, disability, and heath OR icf) AND (life quality OR quality of life): MEDLINE with Full Text, CINHAL PLUS with Full Text, Psychology and Behavioral Sciences Collection, and Academic Search Complete, through the EBSCOhost Research Databases. Titles and abstracts and subsequently full-text articles were screened by two independent reviewers for eligibility, with disagreements resolved by a third reviewer. More details are displayed in Figure 1.

### 2.2. Data Extraction

Data were extracted by one reviewer and cross-checked by a second reviewer. All of the selected papers were codified with one number, the country abbreviation, and date of publication. The quality and risk of bias of the included studies was assessed by two reviewers using The Joanna Briggs Institute Critical Appraisal tools for use in JBI Systematic Reviews—Checklist for Analytical Cross-Sectional Studies [28] (Table 1). The extracted information was summarized to a table, including the year and country where the study was carried out; research question; study design; instruments used to measure the variables; sample characteristics; statistical methods; and results regarding the research question of interest. Extracted data were standardized to present comparable information.

## 3. Results

### 3.1. Literature Search

After title/abstract screening, 13 of 24 studies were included for full-text analysis. Finally, eight (8) publications were included in the final synthesis. From these, one study was excluded after quality and risk of bias were assessed [28].

All studies analyzed the association between anxiety and/or depression with functionality and/or quality of life in a population with a primary chronic condition: lung cancer [LC] and chronic heart disease [CHD] [29]; atopic dermatitis [AD] [30]; chronic obstructive pulmonary disease [COPD] [31]; breast cancer [32]; Parkinson’s disease [PD] [33]; hip and knee osteoarthritis [HKOA] [34]; and critical illness [35].

### 3.2. Study Characteristics

Included in this systematic review were seven (7) studies involving a total of 2279 individuals (range between 60 and 1278), with a mean age ranging from 51.25 ± 16.97 years to 68.28 ± 9.18 years old, and with a sex distribution of 44.44% males and 55.56% females. All included studies are cross-sectional (100%). Two studies (28.57%) took place in Brazil [29,32]; one (14.29%) in the United States [30]; one (14.29%) in Canada [31]; one (14.29%) in Turkey [33]; one (14,29%) in Polonia [34]; and one (14,29%) in Australia [35].

Six (85.71%) studies used the Hospital Anxiety and Depression Scale [HADS] [29,30,31,32,34,35] and one (14,29%) used the Beck Depression Inventory [BDI] and the Beck Anxiety Inventory [BAI] [33,37] to assess depression and anxiety.

Regarding the instruments that were used to evaluate functionality and quality of life, there was not as much consensus as in anxiety and depression, as can be seen in Table 1. In some studies, more than one instrument has even been used to assess functionality. Two (28.57%) studies used the Study 36-item Short-Form Health Survey [SF-36] [29,34]; Chiesa Fuxench et al. (2019) used the Health-Related Quality of Life [HRQoL] and the Dermatology Life Quality Index [DLQI] (14.29%) [30]; Colombino et al. (2020) used the European Organization for Research and Treatment of Cancer [EORTC] QLQ-C30, and its subscale for breast cancer, the EORTC QLQ- BR23 (14.29%) [32]; Aktar et al. (2020) assessed quality of life using 8-item Parkinson’s Disease Questionnaire [PDQ-8] (14,29%) [33]; and Hodgson et al. (2017) used the Health-Related Quality of Life [EQ5D-5L] (14,29%) to assess quality of life [35]. For functionality, three studies used the 6-Minute Walk Distance [6MWD] test [29,31,33]; two studies used the 36-item WHO Disability Assessment Schedule [WHODAS] [34,35]; D’Amore et al. (2022) used the Short Physical Performance Battery [SPPB] and the Late Life Disability Instrument [LLDI] [31]; Colombino et al. (2020) used the Shoulder Pain and Disability Index [SPADI] [32]; Aktar et al. (2020) used the Movement Disorder Society Unified Parkinson’s Disease Rating Scale [MDS-UPDRS], the SenseWear Arm Band activity monitor [SWA], the Timed Up and Go Test [TUG], the Modified Hoehn and Yahr staging scale [YAHR], and the Fatigue Impact Scale [FIS] [33].

Next, we will briefly describe the purpose and results of each study included in the review. More details can be seen in Table 2.

The objectives of Nogueira et al. (2017) were to evaluate the measurement properties of the Identity–Consequence Fatigue Scale [ICFS], as well as the intensity of fatigue and the associated factors (depression, anxiety, and quality of life), in patients with lung cancer [LC] and chronic heart disease [CHD] [29]. They concluded that the ICFS is a valid, reliable instrument for evaluating LC patients, in whom depression and quality of life seem to be significantly associated with fatigue [29]. They also concluded that there was no difference between the LC and CHD groups regarding the ICFS summary variable “*fatigue experiences*” [29]. Both groups presented with a higher level of fatigue when compared with the control group, and there was a significant difference when the control group was compared with the LC and the CHD groups (*p* < 0.001 for both), and when the LC group was compared with the CHD group (*p* < 0.001) [29]. Anxiety and depression scales were significantly correlated with the summary variables “*fatigue experiences*” (r = 0.43; *p* ≤ 0.01, and r = 0.60; *p* ≤ 0.01, respectively) and “*fatigue impacts*” (r = 0.62; *p* ≤ 0.01, and r = 0.63; *p* ≤ 0.01, respectively) [29]. The same summary variables correlated negatively with the mental component summary of SF-36 (r = −0.55; *p* < 0.01, and r = −0.48; *p* ≤ 0.01, respectively) [29].

The objectives of Chiesa Fuxench et al. (2019) were to determine the prevalence of atopic dermatitis [AD] in the population of the United States, along with the distribution of disease severity, and its impact on health-related quality of life [30]. They concluded that patients with AD and those with more severe diseases had higher scores in the Dermatology Life Quality Index and the Hospital Anxiety and Depression Scales vs. control individuals, indicating a worse impact on quality of life and an increased likelihood of anxiety or depression [30]. The percentage of participants with AD who reported a moderate to large effect on quality of life using established thresholds was higher compared with control individuals (24.37 vs. 4.66; *p* < 0.001) [30]. Additionally, 24.73% of participants with AD met clinical criteria for anxiety compared with 9.20% in the control group (*p* < 0.001), and the proportion of participants who met clinical criteria for depression was 13.98% in the AD group versus 5.99% in the control group (*p* < 0.003) [30]. They also concluded that patients with AD were 6 times more likely to report a moderate to severe impact on the DLQI when compared with control individuals (mean = 6.73, 95% CI = 3.41–13.24) [30].

D’Amore et al. (2022) aimed to examine the relationship between International Classification of Functioning, Disability, and Health [ICF] factors commonly measured during pulmonary rehabilitation and participation in life situations among people with chronic obstructive pulmonary disease [COPD] [31]. To this end, they examined the relationships between individual factors and participation, as well as the relationships between different ICF components and participation, and identified the combination of factors (from all ICF components) that explained the most variation in participation [31]. They found that participation in life situations in people with COPD is associated with multiple ICF components; psychological distress (i.e., anxiety and depression symptoms) and mobility were important determinants of participation frequency and limitations [31]. They found that all individual factors were significantly associated with the LLDI limitation scale, a 1-point increase in the HADS resulted in a 0.72-point decrease in the participation limitation score, and a 1-min increase during the 6MWT would result in a 0.05-point increase in the participation limitation score [31].

Colombino et al. (2020) evaluated the impact of a return to work on the quality of life of breast cancer patients and identified factors related to a nonreturn to work [NRTW] [32]. They concluded that breast cancer treatment decreased the women’s work capacity and that a return to work improved the patients’ quality of life [32]. They found that SPADI results for incapacity and pain are lower in people who return to work than in people that could not return (24.05 and 23.87 vs. 42.53 and 52.51, correspondingly). The same happened with the HADS results for anxiety and depression (8.69 and 6.91 vs. 11.37 and 8.70, correspondingly).

The objectives of Aktar et al. (2020) were to compare the effect of biopsychosocial factors based on ICF domains in sedentary and non-sedentary Parkinson’s disease (PD) patients, and to investigate the association between physical activity levels and biopsychosocial factors within sedentary and non-sedentary PD patients [33]. They concluded that patients with a sedentary lifestyle had worse scores in postural control/balance, sit-to-stand, and walking performance [33]. The authors concluded that there is no significant difference between the two groups for the scales BAI (*p* = 0.198), BDI (*p* = 0.988), PDQ-8 (*p* = 0.466), MDS-UPDRS (*p* = 0.188, *p* = 0.642), WA—Step width (cm) (*p* = 0.799), WA—Step length (cm) (*p* = 0.079), YAHR (*p* = 0.235), and FIS (*p* = 0.787). However, they found significance for the 6MWT (*p* = 0.023) and TUG (*p* = 0.003) [33].

Bejer et al. (2021) aimed to examine the psychometric properties of the Polish version of the 36-item WHO Disability Assessment Schedule 2.0 [WHODAS 2.0] in a population with hip and knee osteoarthritis [HKOA] [34]. They concluded that the Polish version of the 36-item WHODAS 2.0 assesses disability according to the ICF in a reliable and valid way [34]. Individuals with HKOA do not present with anxiety or depression on the HADS (HADS-A: mean = 7.09, SD = 3.60; HADS-D: mean = 6.04, SD 4.04) and have low results on the WHODAS 2.0 (WHODAS 2.0 total: mean = 38.33; SD = 17.28) [34]. An increase in the HADS results in an increase in the WHODAS 2.0 total score [34].

Finally, the objective of Hodgson et al. (2017), using patient-reported outcomes, was to measure key components of the World Health Organization’s ICF factors that are relevant to survivors of critical care [35]. In summary, 25% of patients reported no disability, 50% reported a mild disability, and 25% reported a moderate to severe disability on the WHODAS 2.0. At six months after their discharge from the ICU, anxiety was reported in 49 (21%), depression in 41 (17%), and depression and/or anxiety in 53 (22%) survivors. Their health-related quality of life was worse at six months after their critical illness.

These seven studies addressed the aim of this systematic review, i.e., to analyze scientific literature about the association between these variables: quality of life and functionality in depression and anxiety.

All analyzed the quality of life except the COPD study [31]. Two studies used the Study 36-item Short-Form Health Survey [SF-36] [29,34], and the other studies used different instruments [30,32,33,35]. All analyzed functionality, except the AD study [30]. Three studies used the 6-Minute Walk Distance [6MWD] test [29,31,33] and two used the WHO Disability Assessment Schedule [WHODAS] [34,35].

We analyzed the results obtained in relation to anxiety and depression according to the pathologies, as shown in Table 2. We observed higher levels of anxiety and depression with increasing disease severity [30]. The same was true for its impact on quality of life [30,35] and on functionality [29,32,33]. All seven articles reviewed show that the higher functionality (return to work, no sedentary lifestyle, and no fatigue) and quality of life are, the lower the values of anxiety and depression will be.

## 4. Discussion

Although there are already three systematic reviews in the literature, one assessing quality of life and anxiety and depression [19], one assessing depression and functionality [16], and one assessing functionality and anxiety [14], none of them assessed the four variables, which is the aim of this review. In addition, two of them are already old, from 2009 and 2016, so they do not include the studies included in this review and no longer contain the most up-to-date information. Although the one that analyzes quality of life and anxiety and depression is more recent, from 2021, and includes the timeframe that our review covers, it does not include any of the studies mentioned here.

The scientific literature shows the lack of a longitudinal and holistic approach related to mental problems. Meshing the variables of quality of life and functionality in depression and anxiety shows the great diversity in the instruments used to measure quality of life and functionality, but consistency in those used to measure anxiety and depression. Bjelland et al. (2002) reviewed the literature on the validity of the HADS and found that this instrument performs well in assessing symptom severity and in identifying cases of anxiety and depression disorders in somatic, psychiatric, and primary care patients and in the general population [38]. This strengthens the HADS (seven + seven items) as an efficient tool to assess these disorders.

In the studies included in this review, women suffer more depression and anxiety than men, which is in line with already well-known data: women have twice the lifetime rates of depression and most anxiety disorders, which can be explain by hormonal differences, coping styles, and social roles, among other factors [39].

Hodgson et al. (2017) found that a history of depression or anxiety, being separated or divorced, and being discharged to another hospital or rehabilitation facility after the primary hospital admission were more common amongst patients with moderate to severe disability [35]. Thus, it can be said that being in a meaningful affective relationship is a protective factor and helps individuals to have better functionality. The literature supports these findings, as an important risk factor for anxiety and depressive disorders is the termination of a relationship with a close person [40]. The quality of the relationship, which includes the scope of mutual support, is of great importance [40]. Colombino et al. (2020) found that women who returned to work had a higher quality of life, higher scores for body image and sexual function, lower disability and pain scores, and lower anxiety and depression [32]. Consequently, being at work, and therefore being functional at work, is a protective factor for anxiety and depression. Similarly, social support is beneficial to an individual’s mental health and can alleviate their anxiety and depression disturbances [41].

In the study by Nogueira et al. (2017), anxiety, depression, and quality of life seem to be significantly associated with fatigue [29], as patients with variously defined fatigue syndromes have comorbid mood or anxiety disorders [42]. Chiesa Fuxench et al. (2019) found greater anxiety and depression in individuals with AD, and higher levels of anxiety and depression with increasing disease severity [30]. The same was true for quality of life, where there was more impact on quality of life with increasing severity of disease [30]. D’Amore et al. (2022) concluded that people with COPD have an increased risk for mobility limitations, anxiety, and depression, because these individuals have compromised physical functionality and therefore participate less in activities [31]. Aktar et al. (2020) concluded that the deterioration of balance, walking, and sit-to-stand functions may lead to activity limitations in patients with PD at a mild or moderate stage [33].

As the seven studies included people with different pathologies (lung cancer, chronic heart disease, atopic dermatitis, chronic obstructive pulmonary disease, breast cancer, Parkinson’s disease, hip and knee osteoarthritis, and critical illness), the way they assessed quality of life and functionality only allows us to infer a supposed causality between being active (return to work, no sedentary lifestyle, and no fatigue) and having less anxiety and depression. We can also conclude that when levels of anxiety and depression are higher, the greater the severity of chronic conditions.

In the study that presented results for both a control group and disease group, there is evidence of a direct correlation between anxiety and depression and being sick or not [30]. Depression and anxiety are associated with an increased risk of multimorbidity and may be associated with a more rapid accumulation of chronic medical conditions [43]. So, in cases of anxiety and/or depression, it is important to investigate if there is comorbidity and if there exists something that affects functionality or quality of life. The reverse is also relevant.

Regarding the instruments applied in the studies analyzed, there seems to be a consensus that the HADS is a good screening tool for anxiety and depression. For functionality, there is an urgent need for a consensus on the instruments to be used. Bejer et al. (2021) concluded that the Polish version of the 36-item WHODAS 2.0 constitutes a considerable support in clinical practice and in national and international scientific research projects relating to patients with KHOA [34].

Concerning the instruments used for quality of life, it was observed that the two studies that used the Study 36-item Short-Form Health Survey [SF-36] and the European Organization for Research and Treatment of Cancer [EORTC] QLQ-C30 [32,34] did not present the results for the total value, but only the domains. We would like to point out that the Health-Related Quality of Life [EQ5D-5L] instrument used by Hodgson et al. (2017) is a self-assessed, health-related quality of life questionnaire, which measures quality of life on a five-component scale, including mobility, self-care, usual activities, pain/discomfort, and anxiety/depression, and is not specific for assessing quality of life in people with a particular disease, and is therefore used more frequently [35]. It was developed by the EuroQol Group to measure health-related quality of life in 2005 and since its development, it has been increasingly applied in populations with various diseases and has been found to have good reliability and sensitivity [44].

Longitudinal studies must be conducted to know the origins and changes in the relationships or interdependencies of all areas of health, according to the WHO definition.

### Limitations

While this approach is logically justified given the strong interrelation between these aspects, it might be beneficial to explicitly acknowledge the difficulty of integrating all of these elements within a systematic review.

To extract data, it was not always easy to identify the results by groups and instruments due to the different primary conditions included. There were only two studies with control groups and for one of them, the results were not presented, which hindered the inference and discussion of results. Although six of the seven articles used the HADS, it was not possible to perform a meta-analysis, as some results presented the means and others presented the medians, and still others joined groups with chronic pathology and the values for each one were not clear.

As for quality of life and functionality, the multiplicity of the instruments and the low commonality did not facilitate discussion of the results.

## 5. Conclusions

The aim of this systematic review was accomplished by meshing the association between quality of life and functionality in depression and anxiety with a bidirectional effect. While there seems to be a consensus in using the HADS to access anxiety and depression, the same cannot be said for assessing functionality and quality of life. However, even when using different instruments, the results indicate the same association. We found higher levels of anxiety and depression with increasing disease severity. The same was true for its impact on quality of life and on functionality. All of the studies included presented a primary chronic condition and the method used to evaluate quality of life and functionality only suggests a possible link between being active (i.e., returning to work, avoiding a sedentary lifestyle, and reduced fatigue) and lower anxiety and depression; this also indicates that higher levels of anxiety and depression are associated with greater disease severity.

The related literature says that mental health conditions significantly exacerbate the progression of chronic illnesses, reduce treatment adherence, and impair overall patient outcomes. Furthermore, the association between chronic diseases and mental health has a bidirectional nature. This represents a critical issue in healthcare, as it highlights the importance of an integrated approach to managing the physical and psychological aspects of chronic conditions.

The findings of this review are useful for clinical practice, as clinicians should be aware of depressive and/or anxious symptoms in patients with chronic illness with high levels of severity and their impact on the patient’s quality of life and functionality. These findings indicate that the higher functionality and quality of life are, the lower the values of anxiety and depression will be. Quality of life is lower before the development of anxiety and depressive disorders, reduces further at the onset of these disorders, and generally improves with remission to premorbid levels. So, for clinical practice, it is important to assess the disorder (recurrent, relapsing, or chronic), and the quality of life and functionality as well. Members of medical teams should be sensitized to the connections between the psychological, social, and somatic spheres for the manifestations of human functioning. One of the advantages of this article is the authors’ emphasis on the importance of early diagnosis and rehabilitation for improving the quality of life.

It is also important to highlight the difference between anxious and/or depressive symptoms secondary to the existence of a disease and the coexistence of anxious and/or depressive disorders with another disease. When a depressive and/or anxiety disorder is identified, it should always be addressed and measured. If left untreated, it can become chronic and worsen an individual’s quality of life and functionality.

All of the studies included in this review had individuals with different pathologies and studied their association with anxiety, depression, quality of life, and functionality. This highlights the importance of a multidisciplinary approach in patients with a chronic disease, focusing not only on clinical outcomes but also on their overall well-being. Few studies were found on the different types of anxiety and depression disorders and their association with functionality and quality of life. The interplay between anxiety, depression, quality of life, and functionality in chronic diseases provides another very relevant avenue of inquiry, given that patients with a chronic illness frequently suffer from psychiatric comorbidities that can, in turn, adversely affect their quality of life and functionality.

Further research is needed to support the relationship between these variables to draw more concrete conclusions with scientific evidence. These findings may also direct researchers towards a future study about the identification of the determinants of depression and anxiety with respect to the quality of life and functionality of adult populations. As the determinants of health and illness are influenced by life contexts, this research would have to be supported by a longitudinal study that includes individuals in several life transitions and with different health situations.

## Figures and Tables

**Figure 1 healthcare-13-00539-f001:**
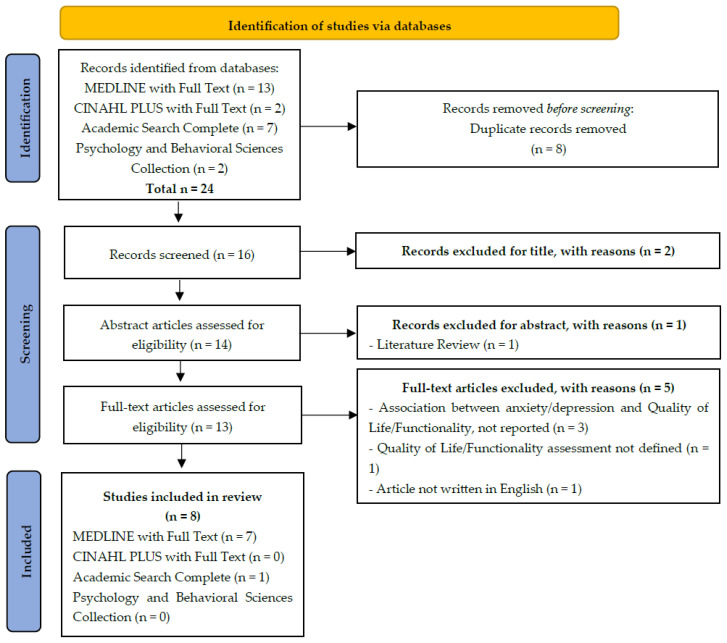
Study flow (PRISMA flow chart).

**Table 1 healthcare-13-00539-t001:** Checklist for analytical cross-sectional studies [28].

Code	Ref.	Quest *								Primary Disease	Anxiety/Depression Instrument	Quality of Life Instrument	Functionality Instrument
		1	2	3	4	5	6	7	8				
3BR2017	[29]	Y	Y	Y	Y	Y	Y	Y	Y	Lung cancer	HADS	SF-36	6MWD
4US2019	[30]	Y	Y	Y	Y	Y	Y	Y	Y	Atopic dermatitis	HADS	HRQoL; DLQI	-
9CA2022	[31]	Y	Y	Y	Y	Y	Y	Y	Y	Chronic obstructive pulmonary disease	HADS	-	6MWD; SPPB; LLDI
10BR2020	[32]	Y	Y	Y	Y	Y	Y	Y	Y	Breast cancer	HADS	EORTC QLQ- BR23	SPADI
11TR2020	[33]	Y	Y	Y	Y	Y	Y	Y	Y	Parkinson’s disease	BDI; BAI	PDQ-8	MDS-UPDRS; SWA; 6MWT; TUG; YAHR; FIS
12PL2021	[34]	Y	U	Y	Y	U	U	Y	Y	Hip and knee osteoarthritis	HADS	SF-36 2.0	WHODAS 2.0
13AU2017	[35]	Y	Y	Y	Y	Y	Y	Y	Y	Critical illness	HADS	EQ5D-5L	WHODAS 2.0 + 3.0
5NEP2021	[36]	Y	Y	N	N	Y	Y	Y	Y	Post-traumatic stress disorder	CIDI 2.1	WHOQOL-BREF	-
%		100	88.89	100	100	77.78	77.78	100	100				
#		9	8	9	9	7	7	9	9				

* 1. Were the criteria for inclusion in the sample clearly defined? 2. Were the study subjects and the setting described in detail? 3. Was the exposure measured in a valid and reliable way? 4. Were objective, standard criteria used for the measurement of the condition? 5. Were the confounding factors identified? 6. Were strategies to deal with the confounding factors stated? 7. Were the outcomes measured in a valid and reliable way? 8. Was an appropriate statistical analysis used? Y = Yes; N = No; U = Unclear; %—Percentage of “Yes” answers. #—Total of answers.

**Table 2 healthcare-13-00539-t002:** The results of each study included in the review.

Code	Ref.	Primary Disease	Anxiety and Depression Values by Groups	Quality of Life Values by Groups	Functionality Values by Groups
3BR2017	[29]	Lung cancer [LC] + Chronic heart disease [CHD]	Lung cancer: - HADS-A: 5 (3–8) - HADS-D: 4.5 (2.0–7.0) *Values are expressed as median (interquartile range).* - No results were presented for control group	Lung cancer: - SF-36 MCSb: 47.7 (13.3) - SF-36 PCSb: 45.6 (8.4) *Values are expressed as mean (SD)* - Fatigue experiences: r = −0.55; *p* < 0.01 - Fatigue impacts: r = −0.48; *p* ≤ 0.01 - No results were presented for control group	- Anxiety fatigue experiences: r = 0.43; *p* ≤ 0.01 - Depression fatigue experiences: r = 0.60; *p* ≤ 0.01 - Anxiety fatigue impacts: r = 0.62; *p* ≤ 0.01 - Depression fatigue impacts: r = 0.63; *p* ≤ 0.01
4US2019	[30]	Atopic dermatitis [AD]	- HADS-A total score: 7.03 (4.80) - HADS-D total score: 5.83 (4.54) *Values are expressed as mean (SD)* - Clear or Mild AD HADS-A total score: 6.09 (4.22) - Moderate AD HADS-A total score: 8.22 (4.81) - Severe AD HADS-A total score: 10.45 (5.22) *Values are expressed as mean (SD)* - Clear or mild AD HADS-D total score: 4.86 (3.99) - Moderate AD HADS-D total score: 6.7 (4.33) - Severe AD HADS-D total score: 8.11 (3.84) *Values are expressed as mean (SD)* - HADS-A control group total score: 4.73 (3.98) - HADS-D control group total score: 3.62 (3.61) *Values are expressed as mean (SD)*	- DLQI total score: 4.71 (6.44) - Clear or mild AD DLQI total score: 2.04 (2.79) - Moderate AD DLQI total score: 5.87 (6.26) - Severe AD DLQI total score: 11.44 (8.48) - DLQI control group total score: 0.97 (2.12) *Values are expressed as mean (SD)*	Ø
9CA2022	[31]	Chronic obstructive pulmonary disease [COPD]	- Every 1-point increase on the HADS was associated with a 0.27 decrease in the participation score - Every 1-point increase in the psychological distress outcome (HADS) resulted in a 0.15 decrease in participation frequency. - No control group	Ø	- One-m increase in 6MWT distance, participation frequency score increased by 0.03 points - One-m increase in 6MWT distance, participation frequency score increased by 0.04 points - No control group
10BR2020	[32]	Breast cancer	- HADS-A total: 9.84 (5.03) - HADS-D total: 7.68 (3.29) *Values are expressed as mean (SD)* HADS-A NRTW: 11.37 (4.91) - HADS-D NRTW: 8.70 (3.00) - HADS-A RTW: 8.69 (4.82) - HADS-D RTW: 6.91 (3.31) *Values are expressed as mean (SD)* - No control group	- Only presents results for each domain and not the total value of the instrument - No control group	- SPADI—incapacity total: 31.99 (26.12) - SPADI—pain total: 41.88 (31.55) - SPADI—incapacity NRTW: 42.53 (27.51) - SPADI—pain NRTW: 52.51 (32.86) - SPADI—incapacity RTW: 24.05 (22.03) - SPADI—pain RTW: 23.87 (28.14) *Values are expressed as mean (SD)* - No control group
12PL2021	[34]	Hip and knee osteoarthritis [HKOA]	- HADS-A: 7.09 (3.60) - HADS-D: 6.04 (4.04) *Values are expressed as mean (SD)* - No control group	- Only presents results for each domain and not the total value of the instrument - No control group	- WHODAS 2.0 total: 38.33 (17.28) *Values are expressed as mean (SD)* - Every 1-point increase in the HADS-A resulted in a 0.475 increase in WHODAS 2.0 total score - every 1-point increase in the HADS-D resulted in a 0.536 increase in WHODAS 2.0 total score
13AU2017	[35]	Critical illness	- No control group	None or mild disability: - EQ5D-5L utility score: 0.77 (0.26) Moderate or severe disability - EQ5D-5L utility score: 0.50 (0.26) *Values are expressed as mean (SD)*	None or mild disability: - HADS-A: 3.50 (3.10) - HADS-D: 2.90 (2.80) Moderate or severe disability - HADS-A: 7.30 (4.70) - HADS-D: 7.00 (4.00) *Values are expressed as mean (SD)*
11TR2020	[33]	Parkinson’s disease [PD]	- BAI sedentary PD: 9.28 (8.73) - BDI sedentary PD: 12.32 (9.41) - BAI non-sedentary PD: 7.60 (10.05) - BDI non-sedentary PD: 11.31 (6.66) *Values are expressed as mean (SD)* *p*-value not significant - No control group	- PDQ-8 sedentary PD: 20.50 (19.51) - PDQ-8 non-sedentary PD: 15.89 (14.97) *Values are expressed as mean (SD)* *p*-value not significant - No control group	- MDS-UPDRS *p*-value not significant - SWA WA- step width (cm) WA- step length (cm) *p*-value not significant for both - WA- walking speed (cm/s). Sedentary PD: 49.84 (17.57) - WA- Walking speed (cm/s). Non-sedentary PD: 62.20 (12.52) - 6MWT sedentary PD: 387.02 (85.16) - 6MWT non-sedentary PD: 445.78 (70.83) - TUG sedentary PD: 9.62 (2.76) - TUG non-sedentary PD: 7.78 (1.20) *Values are expressed as mean (SD)* - YAHR *p*-value not significant - FIS *p*-value not significant - No control group

Ø—Variable not evaluated on that study.

## Data Availability

No new data were created in this study. Data sharing is not applicable to this article.

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
