# Peer review of "Meshing Anxiety, Depression, Quality of Life, and Functionality in Chronic Disease"

_healthcare, 2025, doi:10.3390/healthcare13050539_

Round 1
Reviewer 1 Report
Comments and Suggestions for Authors
This article presents the results of a systematic review of previous analyses of the associations between anxiety, depression, quality of life, functionality and chronic disease. The systematic review reported by the authors was conducted using the Preferred Reporting Items for Systematic (PRISMA) guidelines. The study flow (PRISMA flow chart) is presented graphically in figure 1.
The article is based on seven studies from eight studies identified from the literature review. The extracted data were standardized to present comparable information.
A total of 2279 individuals (aged 60 and 78 years), with a mean age ranging from 51.25 ± 16.97 years to 68.28 ± 9.18 years old, were included. Of these, 44.44% were males and 55.56% were females.
The results of each study are presented in Table 2, broken down by variables: anxiety and depression; quality of life; functionality; chronic disease.
The authors found that one tool is preferred for measuring anxiety and depression (most often it is HADS, less often it is BDI (or other tools)), but different tools are used to measure quality of life and functionality (functioning in various areas of human activity). They refer to self-report, self-assessment. This was one of the factors that made it difficult to formulate conclusions regarding the interdependence of anxiety and depression, quality of life, functionality and chronic disease in people studied in middle age (its second and third phase) and late adulthood.
Based on the systematic review, the authors formulated a conclusion regarding the relationships between mental health, somatic health and social functioning. It is generally consistent with previous findings on the relationships between various aspects of health and the importance of professional activity for the subjective assessment of quality of life.
The results of the systematic review could be discussed by referring to a holistic approach to human health, namely biopsychosocial. The authors consider the longitudinal approach to be justified - for further exploration of the problem of relationships between the above-mentioned variables. It is justified if we want to know the origins and changes in the relationships or interdependencies of all areas of health (health understood according to the WHO definition). We should agree with this suggestion - the systematic review shows a lack of a longitudinal approach. The article is interesting - it may particularly interest members of medical teams and sensitize them to the connections between the psychological, social and somatic spheres for the manifestations of human functioning. One of the advantages of the article is the authors' emphasis on the importance of early diagnosis and rehabilitation for improving the quality of life.
Author Response
Comments 1: This article presents the results of a systematic review of previous analyses of the associations between anxiety, depression, quality of life, functionality and chronic disease. The systematic review reported by the authors was conducted using the Preferred Reporting Items for Systematic (PRISMA) guidelines. The study flow (PRISMA flow chart) is presented graphically in figure 1.
Response 1: Thank you for your expert opinion. Nothing to declare.
Comments 2: The article is based on seven studies from eight studies identified from the literature review. The extracted data were standardized to present comparable information.
Response 2:Thank you for your expert opinion. Nothing to declare.
Comments 3: A total of 2279 individuals (aged 60 and 78 years), with a mean age ranging from 51.25 ± 16.97 years to 68.28 ± 9.18 years old, were included. Of these, 44.44% were males and 55.56% were females.
Response 3: Thank you for your expert opinion. Nothing to declare.
Comments 4: The results of each study are presented in Table 2, broken down by variables: anxiety and depression; quality of life; functionality; chronic disease.
Response 4: Thank you for your expert opinion. Nothing to declare.
Comments 5:
The results of each study are presented in Table 2, broken down by variables: anxiety and depression; quality of life; functionality; chronic disease.
The authors found that one tool is preferred for measuring anxiety and depression (most often it is HADS, less often it is BDI (or other tools)), but different tools are used to measure quality of life and functionality (functioning in various areas of human activity). They refer to self-report, self-assessment. This was one of the factors that made it difficult to formulate conclusions regarding the interdependence of anxiety and depression, quality of life, functionality and chronic disease in people studied in middle age (its second and third phase) and late adulthood.
Response 5: Thank you for your expert opinion. Nothing to declare.
Comments 6: Based on the systematic review, the authors formulated a conclusion regarding the relationships between mental health, somatic health and social functioning. It is generally consistent with previous findings on the relationships between various aspects of health and the importance of professional activity for the subjective assessment of quality of life.
Response 6: Thank you for your expert opinion. Nothing to declare.
Comments 7: The results of the systematic review could be discussed by referring to a holistic approach to human health, namely biopsychosocial. The authors consider the longitudinal approach to be justified - for further exploration of the problem of relationships between the above-mentioned variables. It is justified if we want to know the origins and changes in the relationships or interdependencies of all areas of health (health understood according to the WHO definition). We should agree with this suggestion - the systematic review shows a lack of a longitudinal approach. The article is interesting - it may particularly interest members of medical teams and sensitize them to the connections between the psychological, social and somatic spheres for the manifestations of human functioning. One of the advantages of the article is the authors' emphasis on the importance of early diagnosis and rehabilitation for improving the quality of life.
Response 7: Thank you for your expert opinion. In order to integrate the holistic view we have made some changes, font in red, in the abstract on lines 30 and 32; in the introduction on line 101; in the discussion on lines 293, 294, 361 and 362; in the conclusion on lines 402 to 407.
Reviewer 2 Report
Comments and Suggestions for Authors
The interplay between anxiety, depression, quality of life, and functionality in chronic disease provides another very relevant avenue of inquiry, given that patients with chronic illness frequently suffer from psychiatric comorbidities that can, in turn, adversely affect their quality of life and functionality. The manuscript is well articulated; however, I have a few comments and suggestions:
1) In a systematic review, finding a direct connection between two different elements can already be challenging. This study aims to establish a link among four variables (anxiety, depression, quality of life, and functionality). While this approach is logically justified given the strong interrelation between these aspects, it might be beneficial to explicitly acknowledge the difficulty of integrating all these elements within a systematic review.
2) Discuss and justify, if possible, the differences found between male and female patients. Are there potential biological, psychological, or sociocultural explanations for these variations?
3)The authors state in the discussion: “So, it is important to identify and address anxiety and depression, since they are associated with significant morbidity and mortality.” However, only one of the studies analyzed is cited to support the link between depression, morbidity, and mortality in patients with chronic conditions. This concept should be further elaborated and supported with additional data, or its formulation should be adjusted accordingly.
Author Response
Comments 1: The interplay between anxiety, depression, quality of life, and functionality in chronic disease provides another very relevant avenue of inquiry, given that patients with chronic illness frequently suffer from psychiatric comorbidities that can, in turn, adversely affect their quality of life and functionality. The manuscript is well articulated; however, I have a few comments and suggestions:
Response 1: Thank you for your expert opinion. Nothing to declare.
Comments 2: In a systematic review, finding a direct connection between two different elements can already be challenging. This study aims to establish a link among four variables (anxiety, depression, quality of life, and functionality). While this approach is logically justified given the strong interrelation between these aspects, it might be beneficial to explicitly acknowledge the difficulty of integrating all these elements within a systematic review.
Response 2: Thank you for your expert opinion. To respond to the difficulty listed, we have added a paragraph in red to the limitations, see lines 364 to 366.
Comments 3: Discuss and justify, if possible, the differences found between male and female patients. Are there potential biological, psychological, or sociocultural explanations for these variations?
Response 3: Thank you for your expert opinion. To answer the question, we've added a sentence that answers the biological, psychological and social potential variations, font in red, on lines 303 and 304.
Comments 4: The authors state in the discussion: “So, it is important to identify and address anxiety and depression, since they are associated with significant morbidity and mortality.” However, only one of the studies analyzed is cited to support the link between depression, morbidity, and mortality in patients with chronic conditions. This concept should be further elaborated and supported with additional data, or its formulation should be adjusted accordingly.
Response 4: Thank you for your expert opinion. We agree with your comment, and because it is only supported by one study, it was decided to remove this sentence from the manuscript, and it thus appears in erasure at 398 and 399, in the lines of the conclusion. In addition, we have added a paragraph in the conclusion, font in red, on lines 418 to 421.
Reviewer 3 Report
Comments and Suggestions for Authors
Title: Interesting concept ‘meshing’ utilised in the title. Perhaps ‘association’ as this was outlined line 110 and to be consistent- only a suggestion.
The introduction is clear and introduces the differences between co-morbidity and some connection between the different disorder and link toward disease. Although, this is important more context is needed surrounding the rationale for this association and highlighted need for systematic review- lines 93-111.
Materials and methods section- clear. PRISMA is coherent with search strategy. Study characteristics are well described. The results revealed some interesting finds surrounding scales, diagnostic criteria and disease. The association between disease and mental disorder was well-established throughout and acknowledgement of quality of life, thus related to gender, relationships and support.
Line 319-322- a little more context surrounding COPD and mental disorder needed link with functionality as this is a key theme/ aim of the research- is needed.
Author Response
Comments 1: Title: Interesting concept ‘meshing’ utilised in the title. Perhaps ‘association’ as this was outlined line 110 and to be consistent- only a suggestion.
Response 1: Thank you for your expert opinion. We're going to keep the word “meshing”, but to make it more coherent, we've rewritten the idea, font in red, on lines 107 and 111.
Comments 2: The introduction is clear and introduces the differences between co-morbidity and some connection between the different disorder and link toward disease. Although, this is important more context is needed surrounding the rationale for this association and highlighted need for systematic review- lines 93-111.
Response 2: Thank you for your expert opinion. To respond to your comment, we have made some changes to the introduction, which appear in red, on lines 70, 86, 93, 101, 107 and 111. As a result of these changes, we ended up slightly altering the abstract, in red, on lines 15, 19 and 20.
Comments 3: Materials and methods section- clear. PRISMA is coherent with search strategy. Study characteristics are well described. The results revealed some interesting finds surrounding scales, diagnostic criteria and disease. The association between disease and mental disorder was well-established throughout and acknowledgement of quality of life, thus related to gender, relationships and support.
Response 3: Thank you for your expert opinion. When reviewing this comment, a few words were changed in the methods to improve comprehension, in red, on lines 118 and 121.
Comments 4: Line 319-322- a little more context surrounding COPD and mental disorder needed link with functionality as this is a key theme/ aim of the research- is needed.
Response 4: Thank you for your expert opinion. We have added in the discussion a sentence, in red, lines 325 and 326, to give more context. Additionally, it was also added a paragraph in the conclusion, in red, on lines 418 to 421, explaining the link.
Round 2
Reviewer 2 Report
Comments and Suggestions for Authors
Thank you, the authors have answered my queries